# A Study on Potential Sources of Perineuronal Net-Associated Sema3A in Cerebellar Nuclei Reveals Toxicity of Non-Invasive AAV-Mediated Cre Expression in the Central Nervous System

**DOI:** 10.3390/ijms26020819

**Published:** 2025-01-19

**Authors:** Geoffrey-Alexander Gimenez, Maurits Romijn, Joëlle van den Herik, Wouter Meijer, Ruben Eggers, Barbara Hobo, Chris I. De Zeeuw, Cathrin B. Canto, Joost Verhaagen, Daniela Carulli

**Affiliations:** 1Department of Neuroregeneration, Netherlands Institute for Neuroscience, Royal Netherlands Academy of Arts and Sciences, Meibergdreef 47, 1105 BA Amsterdam, The Netherlands; g.gimenez@nin.knaw.nl (G.-A.G.); m.romijn@nin.knaw.nl (M.R.); j.van.den.herik@nin.knaw.nl (J.v.d.H.); w.meijer@nin.knaw.nl (W.M.); ruben.eggers@knaw.nl (R.E.); b.hobo@nin.knaw.nl (B.H.); j.verhaagen@nin.knaw.nl (J.V.); 2Department of Cerebellar Coordination & Cognition, Netherlands Institute for Neuroscience, Royal Netherlands Academy of Arts and Sciences, Meibergdreef 47, 1105 BA Amsterdam, The Netherlands; c.de.zeeuw@nin.knaw.nl (C.I.D.Z.); c.canto@nin.knaw.nl (C.B.C.); 3Department of Neuroscience, Erasmus MC, 3015 GD Rotterdam, The Netherlands; 4Center for Neurogenomics and Cognitive Research, Amsterdam Neuroscience, Vrije Universiteit Amsterdam, 1081 HZ Amsterdam, The Netherlands

**Keywords:** Semaphorin 3A, perineuronal nets, cerebellar nuclei, Purkinje cells, choroid plexus, AAV-PHP.eB, Cre, toxicity

## Abstract

Semaphorin 3A (Sema3A) is an axon guidance molecule, which is also abundant in the adult central nervous system (CNS), particularly in perineuronal nets (PNNs). PNNs are extracellular matrix structures that restrict plasticity. The cellular sources of Sema3A in PNNs are unknown. Most Sema3A-bearing neurons do not express *Sema3A* mRNA, suggesting that Sema3A may be released from other neurons. Another potential source of Sema3A is the choroid plexus. To identify sources of PNN-associated Sema3A, we focused on the cerebellar nuclei, which contain Sema3A^+^ PNNs. Cerebellar nuclei neurons receive prominent input from Purkinje cells (PCs), which express high levels of *Sema3A* mRNA. By using a non-invasive viral vector approach, we overexpressed Cre in PCs, the choroid plexus, or throughout the CNS of *Sema3A^fl/fl^* mice. Knocking out *Sema3A* in PCs or the choroid plexus was not sufficient to decrease the amount of PNN-associated Sema3A. Alternatively, knocking out *Sema3A* throughout the CNS induced a decrease in PNN-associated Sema3A. However, motor deficits, microgliosis, and neurodegeneration were observed, which were due to Cre toxicity. Our study represents the first attempt to unravel cellular sources of PNN-associated Sema3A and shows that non-invasive viral-mediated Cre expression throughout the CNS could lead to toxicity, complicating the interpretation of Cre-mediated *Sema3A* knock-out.

## 1. Introduction

Semaphorin 3A (Sema3A), a chemorepulsive axon guidance cue during nervous system development [1,2], is also expressed in the adult nervous system [3], where it is mainly found in perineuronal nets (PNNs) [4,5]. PNNs are condensed aggregates of extracellular matrix (ECM) that enwrap the soma, proximal dendrites, and axon initial segment of several types of central nervous system (CNS) neurons. PNNs control synapse function and stabilization (see for review [6,7,8]). Sema3A contributes to the PNN-mediated restriction of visual cortex plasticity [9] and is involved in the regulation of hippocampus-dependent learning [10].

The precise cellular sources of sema3A in PNNs are not known. In several brain regions, there is a mismatch between neurons containing *Sema3A* mRNA and neurons surrounded by a Sema3A^+^ PNN. For instance, Sema3A is found in PNNs around neocortical interneurons mainly in layers IV, V, and VI [4], whereas *Sema3A* mRNA is detected in neurons in layers II–III [3]. In addition, *Sema3A* mRNA is highly expressed in Purkinje cells (PCs) [3,11] and Sema3A protein is abundant in PNNs around cerebellar nuclei neurons [11], which receive prominent innervation from PCs [12]. After the partial depletion of PCs, levels of Sema3A in the cerebellar nuclei decrease [11]. These observations suggest that Sema3A may derive from neurons that secrete Sema3A from their axons into the extracellular environment. Indeed, in vitro experiments demonstrated that Sema3A is released by axons and dendrites in an activity-dependent manner and captured in the ECM [13]. Besides neurons, potential sources of Sema3A are the choroid plexus (ChP), the meninges, and endothelial cells in the median eminence and the pineal gland, as they express *Sema3A* mRNA [3,14,15]. After release from the ChP, Sema3A may be transported via the cerebrospinal fluid to the whole CNS, where it would bind to PNNs, similarly to what occurs to another PNN component, orthodenticle homeobox 2 (Otx2) [16].

To elucidate the main cellular contributors to PNN-associated Sema3A, we focused on the cerebellar nuclei, whose large projection neurons are enwrapped by Sema3A^+^ PNNs [4,11]. Here, we investigated whether Sema3A protein expression levels change following *Sema3A* knock-out (KO), specifically in PCs or the ChP, by employing a Cre-loxP approach. To target PCs, we injected *Sema3A^fl/fl^* mice intravenously with an adeno-associated virus (AAV) variant that can cross the blood–brain barrier (AAV-PHP.eB) [17] overexpressing Cre under the PC-specific promoter L7 [18]. To target the ChP, we injected AAV5-CAG-Cre in the lateral ventricles of *Sema3A^fl/fl^* mice. Because these approaches did not result in a decrease in Sema3A in the cerebellar nuclei, we knocked out *Sema3A* simultaneously in the ChP, in PCs, and in neurons throughout the CNS by expressing Cre via an AAV-PHP.eB with a synapsin (SYN) promoter [19]. This global CNS approach was successful in decreasing PNN-associated Sema3A. However, global knock-out (KO) mice exhibited motor deficits, as well as neurodegeneration and microglia reactivity throughout the CNS. Because a growing amount of evidence demonstrates the toxic effects of Cre in the brain and other organs [20,21,22,23,24,25,26], which are likely due to Cre-induced DNA damage at pseudo-loxP sites [27], we investigated whether the observed phenotype was linked to Cre toxicity. Following the injection of AAV-PHP.eB-L7-Cre, AAV-PHP.eB-SYN-CreGFP, and AAV5-CAG-Cre in wild-type (WT) mice, we observed the same behavioral and anatomical outcomes as in *Sema3A^fl/fl^* mice, indicating that Cre expression throughout the CNS in adult mice exerts toxic effects.

## 2. Results

### 2.1. Knocking-Out Sema3A in Purkinje Cells Does Not Decrease Sema3A Content in PNNs Around Cerebellar Nuclei Neurons

*Sema3A* mRNA is abundantly expressed in adult PCs. This led us to hypothesize that Sema3A, axonally transported by PCs, is a major source of Sema3A in PNNs of their target neurons in the cerebellar nuclei. To elucidate the contribution of PCs to Sema3A accumulation in PNNs in the cerebellar nuclei, we knocked out *Sema3A* specifically in PCs via a Cre-loxP system, namely by intravenously injecting *Sema3A^fl/fl^* mice with AAV-PHP.eB expressing Cre under a PC-specific promoter (L7). Control mice were *Sema3A^fl/fl^* mice injected with AAV-PHP.eB-L7-GFP. Three weeks post-injection, strong GFP or Cre expression was detected in the majority of PCs in both the vermis and the cerebellar hemispheres. GFP was apparent in all PC compartments (cell body, dendrites, and axons), whereas Cre exhibited a nuclear localization (Figure 1A,B,D,E). On average, 88% of PCs in the vermis and 84% of PCs in the hemisphere expressed GFP (Figure 1C). To check whether the PC-specific Cre-loxP approach induces Cre-mediated excision of *Sema3A* exon 1 in PCs, we performed a PCR for *Sema3A* on DNA extracted from PCs. A band of 1.6 kb, corresponding to the intact *Sema3A* gene, was detected in L7-GFP, L7-Cre, or L7-CreGFP mice. In addition, a band of around 500 bp was detected in L7-Cre and L7-CreGFP but not in the L7-GFP PCs (Figure 1F), indicating the correct excision of *Sema3A* exon 1 in PCs following Cre recombination. Because the lifespan of extracellular matrix proteins can reach up to 12 weeks [28], we quantified the Sema3A content in PNNs around neurons in the anterior part (IntA) and the posterior part (IntP) of the cerebellar interpositus nucleus at 12 weeks post-injection. Unexpectedly, we found an increase in the intensity of PNN-associated Sema3A in both the IntA and IntP of PC KO mice (Figure 1G–L; Mann–Whitney U test, IntA *p* = 0.0035, IntP *p* < 0.0001). These data show that knocking out *Sema3A* only in PCs does not induce a decrease in Sema3A protein levels in PNNs in the cerebellar nuclei but leads to an increased expression, suggesting that other cellular sources of Sema3A may overcompensate for the decreased Sema3A production from PCs.

### 2.2. Knocking-Out Sema3A in the Choroid Plexus Does Not Decrease Sema3A Content in PNNs of Cerebellar Nuclei Neurons

The ChP is a source of the PNN component Otx2 [16]. After being secreted by the ChP, Otx2 is transported via the cerebrospinal fluid to the brain parenchyma, where it is captured by PNNs [16]. Because the ChP expresses *Sema3A* mRNA [14] and we observed strong Sema3A immunolabelling in the ChP (Figure 2A), we hypothesized that, similarly to Otx2, Sema3A in PNNs may derive from the ChP. We aimed to knock-out *Sema3A* in the ChP by overexpressing Cre specifically in the ChP of *Sema3A^fl/fl^* mice. Chen et al. (2020) observed effective transduction of the ChP by AAV1 or AAV5 containing a CAG or CMV promoter, respectively [29]. To compare the transduction efficiency of these AAVs in the ChP, we injected AAV1-CAG-GFP or AAV5-CAG-GFP in the lateral ventricles of C57BL/6J mice. Three weeks after injection, both viral vectors efficiently transduced the ChP (Figure 2B; Appendix A). When using AAV5, the percentage of Sema3A^+^ ChP cells expressing GFP was ~63% in the lateral ventricle and ~54% in the fourth ventricle (Figure 2C), whereas when using AAV1, the percentage of Sema3A^+^ ChP cells expressing GFP was slightly lower (~46% in the lateral ventricle and ~49% in the 4th ventricle; Appendix A). In addition, in AAV1 or AAV5 injected mice, we found that the majority of GFP^+^ ChP cells were Sema3A^+^, namely ~86% in the lateral ventricles and ~96% in the 4th ventricle with the AAV5 capsid (Figure 2D) and ~88% in the lateral ventricles and ~86% in the 4th ventricle with the AAV1 capsid (Appendix A). To investigate whether the ChP is a source of Sema3A in PNNs of cerebellar nuclei neurons, we injected *Sema3A^fl/fl^* mice with AAV5-CAG-Cre (ChP KO) or AAV5-CAG-GFP virus (ChP GFP) and analyzed the Sema3A content in PNNs around neurons in the IntA and IntP 10 weeks after injection. Similar to what was seen 3 weeks after injection, a strong expression of GFP or Cre was observed throughout the ChP in the lateral (Figure 2E–G) and fourth ventricles (Figure 2H–J), with minimal transduction of the parenchyma. A PCR on DNA extracted from the ChP demonstrated a correct excision of *Sema3A* exon 1 by Cre recombination in ChP KO mice (Figure 2K). When we quantified the Sema3A levels around cerebellar nuclei neurons, we found an increase in Sema3A content in PNNs around IntA and IntP neurons in ChP KO mice when compared to ChP GFP mice (Figure 2L–Q; Mann–Whitney U test, IntA *p* = 0.0351, IntP *p* = 0.0011). These data show that, similarly to what occurs in the PC KO model, knocking out *Sema3A* only in the ChP results in an increase in PNN-associated Sema3A protein levels in the cerebellar nuclei, suggesting that other cellular sources of Sema3A may overcompensate for the decreased Sema3A production from the ChP.

### 2.3. Globally Knocking-Out Sema3A Decreases Sema3A Content in PNNs

Because the amount of Sema3A in PNNs of cerebellar nuclei neurons is not decreased by targeting only PCs or ChP cells, we studied the effects of knocking out *Sema3A* in multiple sources simultaneously, including PCs, ChP, the cerebellar nuclei themselves, and neurons throughout the brain. To this end, we adopted a global CNS knock-out strategy. Namely, *Sema3A^fl/fl^* mice were injected intracerebroventricularly with AAV5-CAG-Cre (to target the ChP) and intravenously with AAV-PHP.eB-L7-Cre (to target PCs) and AAV-PHP.eB-SYN-CreGFP (to target other CNS neurons)—these mice will be referred to as global KO mice. Control mice were *Sema3A^fl/fl^* mice injected with AAV5-CAG-GFP, AAV-PHP.eB-L7-GFP, and AAV-PHP.eB-SYN-GFP, referred to as global GFP mice. Approximately 10 weeks after injections, we observed a strong GFP or Cre transduction in PCs, in ChP cells, and in neurons throughout the brain, including the spinal cord, brainstem, cerebellar nuclei, and several cortical areas. Example pictures of GFP^+^ or Cre^+^ neurons in the Int and the primary somatosensory cortex barrel field (S1bf) are shown in Figure 3A,B and Figure 3C,D, respectively. Cre recombination driven by the SYN-CreGFP construct was checked by extracting DNA from the somatosensory cortex of global KO mice. The PCR showed a correct excision of *Sema3A* exon 1 (Figure 3E). To investigate whether a global KO of *Sema3A* results in a decrease in Sema3A in PNNs, we quantified Sema3A content in PNNs around neurons in the IntA and IntP. Here, we found a significant decrease in Sema3A^+^ PNN intensity in global KO mice when compared to global GFP mice (Figure 3F–K; Mann–Whitney U test, IntA *p* < 0.0001, IntP *p* < 0.0001). Next, to assess whether Sema3A was generally decreased in the CNS, we evaluated Sema3A levels in the S1bf. Similarly to what we found in the Int, we observed a significant decrease in PNN-associated Sema3A expression in the S1bf in global KO mice (Figure 3L–N; Mann–Whitney U test, *p* = 0.0038). These data suggest that a knock-out approach targeting multiple cellular sources of Sema3A is effective in reducing Sema3A protein in PNNs.

### 2.4. Cre Expression Throughout the CNS Leads to Motor Deficits

Interestingly, global KO mice exhibited motor deficits, such as limping, hind paw dragging, and reduced balance while walking. Moreover, they showed hindlimb clasping when grabbed by the tail (Appendix A). These motor abnormalities were not observed in global GFP mice. When we compared motor balance and the coordination of global KO mice and global GFP mice by testing them on the rotarod for three consecutive days, we found that global KO mice performed significantly worse than control mice (Figure 4; 2way ANOVA with repeated measures, Conditions *p* = 0.0001, F(1,8) = 48.05; Šidák multiple comparisons test, day 1 *p* = 0.0005, day 2 *p* < 0.0001, day 3 *p* < 0.0001). A growing body of evidence points to Cre toxicity in the brain, affecting mouse behavior [26,30,31]. Therefore, we investigated whether Cre overexpression driven by our global viral vector approach is accountable for the observed motor phenotype. We injected WT mice (namely CD-1 mice, the background strain of *Sema3A^fl/fl^* mice) with the same AAV vectors expressing Cre that were used to obtain global KO mice (these mice will be referred to as CD-1 Cre). Uninjected CD-1 mice were used as control mice. Two to three weeks post-injection, strong motor deficits were observed in four out of five CD-1 Cre mice already. They showed motor impairments that were generally more exacerbated than in global KO mice, with some mice not being able to walk and affected by limb muscle spasms (Appendix A). The fifth CD-1 Cre mouse started to show signs of motor deficits in the fourth week post-injection, which were similar to those of global KO mice, i.e., limping, hindpaw dragging, and reduced balance while walking. Because the motor impairment in the majority of CD-1 Cre mice was severe, we could not perform the rotarod test on these mice. Overall, our findings highlight that Cre expression throughout the whole CNS can affect motor functions in adult mice.

### 2.5. Cre Expression Throughout the CNS Leads to Neuroinflammation and Neurodegeneration

To shed light onto the cause of the observed motor symptoms, we investigated whether there were signs of neuroinflammation and neurodegeneration in motor-related CNS areas of mice globally expressing Cre. Abnormal neuroinflammation and neurodegeneration have been previously reported in association with Cre expression in the substantia nigra pars compacta (SNc) [26,32] and other brain regions following local AAV injections [31,33] but no data exist about Cre toxicity in the CNS following systemic AAV administration. First, we examined whether microglia reactivity occurs in the ventral horn of the spinal cord (SC), the Int and the SNc of global KO mice, and CD-1 Cre-injected mice. Microglia were detected by using anti-Iba1 antibodies. Strong microglia reactivity was apparent in the SC and Int of Cre-injected mice, namely, a 2-fold increase in Iba1 intensity was found in global KO mice when compared to global GFP mice (Figure 5A,B,E,F,M; unpaired *t*-test, SC *p* = 0.0113, t = 3.176, Int *p* = 0.0076, t = 3.335) and a 4-fold increase was found in CD-1 Cre mice when compared to CD-1-uninjected mice (Figure 5C,D,G,H,N; unpaired *t*-test, SC *p* = 0.0066, t = 4.47, Int *p* = 0.0053, t = 3.989). Whereas no change in Iba1 expression levels was observed in the SNc of global KO mice (Figure 5I,J,M; unpaired *t*-test, *p* = 0.83, t = 0.22), strong microglia activation was found in the SNc of CD-1 Cre mice (Figure 5K,L,N; unpaired *t*-test, *p* = 0.0026, t = 4.568). This was accompanied by a higher amount of Cre^+^ cells throughout the CNS in CD-1 Cre mice when compared to global KO mice. Example pictures of widespread Cre expression in the CNS of global KO mice and CD-1 Cre mice are shown in Appendix A, respectively. Overall, strong microglia reactivity throughout the CNS was observed in Cre-injected mice. Representative pictures of the widespread inflammation in the CNS of CD-1 Cre mice compared to CD-1-uninjected mice are shown in Appendix A. Motor areas (e.g., primary motor cortex and substantia nigra; blue arrows), as well as non-motor areas (e.g., hippocampus, visual cortex, principal sensory trigeminal nucleus; white arrows), present signs of inflammation (Appendix A).

To investigate whether neurodegeneration occurs in Cre overexpressing mice, we quantified the density of neurons (number/mm^2^) in the SC, Int, and SNc of global KO mice. We used NeuN antibodies to stain neurons in the SC and the Int and anti-tyrosine hydroxylase (TH) antibodies to stain neurons in the SNc. We did not find any change in NeuN^+^ neuronal density in the SC of global KO mice when compared to global GFP mice (Figure 6A–C; unpaired *t*-test, *p* = 0.92, t = 0.11) despite the presence of several Cre-expressing neurons (Appendix A). Similarly, we did not observe any change in the density of TH^+^ neurons in the SNc (Figure 6D–F; unpaired *t*-test, *p* = 0.31, t = 1.07), where a sparse Cre transduction was found (Appendix A). However, the density of neurons was decreased in the Int of global KO mice (~28% decrease; Figure 6G–I; unpaired *t*-test, *p* = 0.0057, t = 3.501). To check whether the neuronal loss was due to Cre expression, we quantified neuronal density in the Int of CD-1 Cre mice. Here, we found a substantial decrease in number of neurons, which was even more pronounced than in global Cre mice (~88% decrease; Figure 6J–L; unpaired *t*-test, *p* = 0.0002, t = 6.944).

These data suggest that motor deficits following a global overexpression of Cre throughout the CNS may be a result of neuroinflammation and degeneration in motor-related CNS areas caused by Cre toxicity.

## 3. Discussion

In this study, we show that (i) knocking-out Sema3A in PCs or the ChP does not induce a decrease in Sema3A levels around cerebellar nuclei neurons but results in an increase; (ii) knocking-out Sema3A in neurons throughout the brain, including PCs, and in the ChP induces a decrease in Sema3A around cerebellar nuclei neurons and cortical neurons; and (iii) AAV-PHP.eB-mediated non-invasive overexpression of Cre throughout the CNS results in motor deficits and induces microglia reactivity and neurodegeneration.

### 3.1. Sources of Sema3A

Previous studies show that *Sema3A* mRNA is expressed in multiple cell types, including neurons [3,11], the ChP [14], endothelial cells [15], and the meninges [3]. In cultured cortical neurons, Sema3A is transported in secretory vesicles along dendrites and axons, which, in the latter, move in the anterograde direction. After depolarization, Sema3A vesicles in axons become more stationary, in association with an increased secretion of Sema3A [13]. Thus, in vivo, neuron-derived Sema3A may be released from axons and then may be incorporated in PNNs in regions situated at a distance from the cell body that synthesizes it. This is in line with the observation that in the cortex *Sema3A* transcripts are detected in pyramidal cells in layers II–III [3], while Sema3A^+^ PNNs are mainly apparent around parvalbumin^+^ neurons scattered throughout all cortical layers [4], which receive input from pyramidal neurons [34]. In the cerebellum, Sema3A immunoreactivity in PNNs around cerebellar nuclei neurons is reduced following partial depletion of PCs (which express strong *Sema3A* mRNA levels), suggesting that Sema3A in cerebellar nuclei may derive from PCs [11]. However, following *Sema3A* knock-out in PCs, we did not find a decrease in Sema3A protein around cerebellar nuclei neurons but an increase. This may suggest that PC-derived Sema3A is decreased but Sema3A from other cellular sources is captured in the PNNs of cerebellar nuclei neurons, resulting in an increase in Sema3A levels in their PNNs. A similar increase around cerebellar nuclei neurons has been observed after knocking out *Sema3A* in the ChP. The ChP is a source of ECM molecules, including proteins of the semaphorin3 family [35] and Otx2 [16]. After being released into the cerebrospinal fluid Otx2 reaches the brain parenchyma where it is captured by PNNs [16]. Although Sema3A may be released from the ChP, our data suggest that the ChP is not the predominant source of PNN-associated Sema3A. Indeed, only when we knocked out *Sema3A* in both the ChP and neurons throughout the brain, including PCs, did we obtain a decrease in Sema3A levels in PNNs in the cerebellar nuclei and the S1bf, suggesting that both cell types may contribute to the presence of Sema3A in PNNs throughout the CNS. Further experiments need to be conducted in order to unravel the precise sources of PNN-associated Sema3A. For instance, it would be interesting to investigate how selectively knocking out *Sema3A* in neurons, excluding PCs, affects Sema3A levels in PNNs in the CNS. In addition, according to the mouse single-cell RNA seq dataset of the Allen brain atlas (https://portal.brain-map.org/atlases-and-data/rnaseq accessed on 1 December 2024), Sema3A is mainly expressed in somatostatin^+^ neurons and glutamatergic neurons. This diversity in neuronal types emphasizes the need to examine their specific roles in Sema3A production.

### 3.2. Cre Toxicity

The investigation of gene function in vivo has, in general, progressed rapidly thanks to the Cre-loxP technology. The phage P1-derived Cre recombinase catalyzes the recombination between loxP recognition sequences and the excision of the DNA segment between loxP sites occurs. Because loxP sites are engineered around crucial exons, this technology results in a loss of gene function. By placing Cre under the control of a region-selective promoter, temporally and spatially controlled gene ablation is obtained. Cre-loxP knock-out can be achieved by using Cre-transgenic mice mated to mice engineered with loxP sites or by viral delivery of Cre to the target organ of mice with loxP sites. Further temporal control of Cre expression has been obtained by fusing Cre to a mutated ligand binding domain of the human estrogen receptor, resulting in tamoxifen-dependent Cre expression [36,37,38].

Since the early 2000s, a growing number of studies have shown that Cre recombinase can have toxic effects in vitro [39,40] and in vivo, including in the brain [21,26,30,31,33,41,42], heart [22,25], hematopoietic cells [43], and spermatids [20]. Cre toxicity is related to the presence of pseudo-loxP sites in the mammalian genome, which can serve as substrates for Cre recombinase [20,27,39,42,44]. DNA damage due to double-strand breaks and nicks is repaired by non-homologous end joining, which can result in chromosome rearrangements [39,45,46]. Cre-induced DNA damage has been proposed to induce programmed cell death either by p53-mediated apoptosis or by autophagy [26,47,48]. This secondary effect of Cre has been used as a tool to selectively ablate certain cell types [47,49].

Approaches to diminish the toxic effect of Cre have been investigated. In vitro assays using self-deleting retroviruses, including lentiviruses, showed an efficient reduction in Cre toxicity [21,40,50]. Alternatively, mutants of Cre were developed to improve its accuracy of recombination, decreasing its binding to pseudo-loxP sites [51]. Interestingly, studies demonstrated that Cre toxicity depends on Cre expression levels [39,46] and that reducing the titer of viral constructs used to overexpress Cre can decrease Cre toxicity [21,32,33,39]. In the majority of transgenic mice expressing Cre under specific cellular promoters, there are no reports of Cre toxicity, suggesting that expression levels of Cre in transgenic mice may be lower than with virally administered Cre. However, Cre toxicity was observed in some Cre transgenic mice. For example, in nestin-Cre mice, developmental Cre toxicity was reported, with a severe reduction in brain parenchyma and hydrocephaly [23,42]. Moreover, neuroinflammation and aberrant synaptic changes occur in the hippocampus of CaMKIIα-iCre mice as well as nestin-Cre mice [41].

To our knowledge, here, we show, for the first time, that the systemic administration of Cre in the CNS by using the AAV-PHP.eB technology results in toxic effects, including microglia reactivity and neurodegeneration. In line with previous evidence showing a dose-dependent Cre toxicity, the intensity of Iba1 expression and the amount of cell degeneration were higher in mice in which Cre expression was stronger (CD-1 mice) than in mice where Cre expression was weaker (*Sema3A^fl/fl^* mice). Although the virus titers were comparable in the two experiments, the virus batches used were different, likely accounting for different transduction levels. Indeed, disparate and variable effects of AAV-Cre viruses are reported, potentially due to differences in specific batches of viruses [31].

We found that mice globally expressing Cre in the CNS exhibit severe motor impairments, as shown by a strongly reduced latency to fall in the rotarod test. However, in transgenic mice overexpressing Cre under the SYN promoter, no motor deficits on the rotarod were detected [30]. This suggests that the expression of Cre in neurons involved in the control of motor behavior may be higher in mice injected with AAV-PHP.eB-SYN-CreGFP than in SYNCre transgenic mice. Alternatively, the simultaneous expression of Cre in neurons and ChP in our global KO approach may account for Cre toxicity in motor circuits. Because we found strong microglia activation in CNS motor areas and neuronal loss in the cerebellum, we can speculate that the observed motor defects are due to these phenomena, although we cannot exclude that dysfunctions in CNS areas controlling sensory feedback contribute to motor impairments. Nonetheless, behavioral deficits as a consequence of Cre toxicity are also described in SYNCre transgenic mice, particularly in males, which exhibit increased anxiety and impaired learning [30].

These findings represent the first attempt to unravel the cellular source(s) of Sema3A in PNNs. We found that knocking out *Sema3A* production in PCs or ChP is not sufficient to decrease the amount of PNN-associated Sema3A in cerebellar nuclei. When knocking out *Sema3A* in ChP and neurons throughout the CNS, we found a decrease in the amount of PNN-associated Sema3A in the CNS. Although the decline in PNN-associated Sema3A may be the result of globally knocking out *Sema3A*, it was accompanied by Cre toxicity. This outcome complicates the interpretation of Cre-mediated *Sema3A* knock-out since we cannot exclude that the adverse effects of the brain-wide Cre-expression contribute to this decline.

## 4. Materials and Methods

### 4.1. Animals

All experimental procedures involving animals were approved by the animal committee of the Royal Netherlands Academy of Arts and Sciences and adhered to the European guidelines for the care and use of laboratory animals (Council Directive 86/6009/EEC). Male and female *Sema3A^fl/fl^* mice (N = 32), CD-1 mice (Charles Rivers, Wilmington, MA, USA) (N = 9), and C57BL/6J mice (Janvier, Genest-Saint-Isle, France) (N = 7) of 2 to 4 months of age were used for this study. The *Sema3A^fl/fl^* mouse line was provided by Prof. Taniguchi, Sapporo Medical University, Japan [52], and bred in-house. Mice were socially housed with ad libitum access to food and water and were kept on a normal light cycle (12:12 light/dark). Experiments were performed in the daytime.

### 4.2. Viral Vector Production

Adeno-associated viral vectors were produced following a protocol previously described [53]. To target PCs, an AAV plasmid with a PC-specific promoter L7 driving the expression of GFP (Addgene #49140) or Cre (Addgene #49117) was packaged into an AAV-PHP.eB capsid [17]. An additional plasmid was cloned in which the Cre sequence was replaced by a CreGFP sequence. This construct was packaged into an AAV-PHP.eB capsid but was only used for the quantification of Sema3A^+^ PNNs of Figure 1 because of low production titers, mostly due to the size of the plasmid (~4.9 kb including ITRs), making it difficult to encapsulate itself in the AAV-PHP.eB capsid. To target other neuronal sources, an AAV plasmid with a synapsin promoter driving the expression of GFP (gifted by Prof. Deniz Kirik, Lund University, Lund, Sweden) or CreGFP (cloned in house by replacing the GFP sequence with a CreGFP sequence) was packaged into an AAV-PHP.eB capsid. To target the ChP, an AAV plasmid with a cytomegalovirus immediate-early enhancer/chicken β-actin/rabbit beta-globin (CAG) promoter driving the expression of GFP was packaged into an AAV1 or AAV5 capsid. These capsids were selected based on their ability to transduce the ChP, as described by Chen et al. (2020) [29]. After testing the transduction efficiency of these 2 viruses, we packaged an AAV plasmid with a CAG promoter driving the expression of Cre (Addgene #51904) into an AAV5 capsid.

### 4.3. Intravenous AAV Delivery

Intravenous AAV injections were performed by placing mice in a restrainer and injecting 100 μL of AAV-PHP.eB-L7-GFP (1.6E^12^ genomic copies (GC)/mouse) or AAV-PHP.eB-L7-CreGFP (1.6E^12^ GC/mouse) in the tail vein (PC KO group) or injecting 150 μL of a mix of AAV-PHP.eB-L7-GFP (1.2E^12^ GC/mouse) and AAV-PHP.eB-SYN-GFP (1.2E^12^ GC/mouse) or AAV-PHP.eB-L7-Cre (1.2E^12^ GC/mouse) and AAV-PHP.eB-SYN-CreGFP (1.2E^12^ GC/mouse) in the retro-orbital venous sinus (global KO group and CD-1 group) under isoflurane anesthesia (induction 4–5%, maintenance 1–2% in medicinal air).

### 4.4. Intracerebroventricular AAV Delivery

Local AAV injections were performed in the lateral ventricles to target the ChP (ChP KO group, global KO group, and CD-1 group). Mice received a subcutaneous injection of carprofen (Rimadyl^®^, Zoetis, Parsippany-Troy Hills, NJ, USA) (5 mg/kg) and another one of buprenorphine (Bupaq^®^, Vetviva, Wels, Austria) (0.1 mg/kg) pre-operative. Mice were anesthetized using isoflurane anesthesia (induction 4–5%, maintenance 1–2% in medicinal air), had their heads shaved, and were placed on a stereotaxic apparatus with a heating pad and a rectal probe to keep the body temperature at around 37 °C. The eyes were covered with eye cream and a piece of paper to protect them from light and prevent corneal desiccation. The head skin was cleaned with hibicet and locally anesthetized with a lidocaine spray (Xylocaine^®^ spray, AstraZeneca, Cambridge, UK) (100 mg/mL). A 1 cm longitudinal skin incision was made on the skull. Lidocaine cream (Xylocaine^®^ cream, AstraZeneca, Cambridge, UK) (50 mg/g) was spread on the incision to locally anesthetize the periosteum. Two small craniotomies were drilled bilaterally and 3.5 μL of AAV1-CAG-GFP or AAV5-CAG-GFP (both 1.1E^11^GC/mouse) or 2.5 μL AAV5-CAG-Cre (1.4E^11^GC/mouse) were injected in each lateral ventricle (A/P: −0.46 mm, Lat: ±1.1 mm, D/V: −2 mm) by using a quartz capillary pipette (30–40 μm tip diameter) connected to a Harvard injection pump at a rate of 0.3 μL/min. The capillary was left in place for 5 min after the injection and then slowly retracted. The head skin was closed using resorbable sutures or Michel clips. Animals were placed in a heated chamber until they recovered from anesthesia. The day after surgery, drinking water with 0.06 mg/mL of carprofen was given to the mice for 3 days.

### 4.5. Rotarod Test

Global GFP (N = 6) and KO mice (N = 4) were tested for motor coordination and balance by using an accelerating rotarod (Ugo Basile, model 47600). Mice were placed on the rotating bar and given a habituation period of 1 min at constant velocity (4 rpm), after which the rotating bar accelerated from 4 to 60 rpm over a period of 300 s. Mice were given 4 trials/day for 3 consecutive days. The latency to fall was recorded.

### 4.6. DNA Extraction and PCR

The efficacy of Cre recombination in *Sema3A^fl/fl^* mice was tested by dissecting out the PC layer from cerebellar slices of a perfused PC KO mouse, the ChP from forebrain and cerebellar slices of a perfused ChP KO mouse, and the neocortex from forebrain slices of a perfused global KO mouse. The DNA was recovered using a RecoverAll^TM^ kit (#AM1975, ThermoFisher, Waltham, MA, USA) and a PCR was performed using a GoTaq^®^ Flexi DNA polymerase kit (Promega, Madison, WI, USA, #M7805) in the following conditions: 95 °C for 2 min (1 cycle), 95 °C for 30 s, 62 °C for 30 s, 72 °C for 2 min (35 cycles), and 72 °C for 7 min. The primer sequences are Forward: GTTCTGCTCCCGGCTCTAATCTC; Reverse: ATGGTTCTGATAGGTGAGGCATGG. PCR products were run in a 1% agarose gel and visualized with 0.01% Ethidium Bromide.

### 4.7. Histological Procedures

Mice were anesthetized via an intraperitoneal injection of an overdose of pentobarbital (Euthasol^®^, Apotheek Universiteit Utrecht, Utrecht, The Netherlands) (500 mg/mL) and transcardially perfused with ~30 mL of freshly prepared phosphate buffer saline (PBS) followed by ~85 mL of freshly prepared 4% paraformaldehyde (PFA) in phosphate buffer (PB, 0.1 M). Brains and spinal cords were post-fixated overnight in the same PFA solution and cryoprotected in 0.1 M PB containing 30% sucrose at 4 °C until they sank. If the tissue was not sectioned within 2 weeks, the solution was replaced by a 30% sucrose solution with 0.02% Na-azide for longer preservation. Brains were cut in 25 µm-thick sagittal or coronal floating slices and the cervical part of the spinal cord was cut transversally in 40 µm-thick floating slices by means of a freezing sledge microtome (Hyrax S30, Zeiss, Oberkochen, Germany) or a cryostat (CM3050S, Leica, Wetzlar, germany). Slices were stored in a PBS solution with 0.02% Na-azide and kept at 4 °C for long-term storage.

All stainings except Sema3A staining were performed according to the following procedure. Slices were incubated overnight at 4 °C with primary antibodies and 5% fetal calf serum (FCS) diluted in PBS with 0.25% Triton X-100 (PBST). Slices were washed 3 times at RT with PBST (10 min/wash) and then incubated for 2 h at room temperature (RT) with secondary antibodies and 2.5% FCS diluted in PBST, followed by 3 washes at RT with PBST (10 min/wash). Finally, slices were incubated for 5 min at RT with DAPI (1:10,000, #D9542, Sigma-Aldrich, Merck, Darmstadt, Germany,) diluted in PBST, followed by three 10 min-washes in PBST. The primary antibodies used were chicken anti-GFP (1:1000, #AB16901, Millipore, Merck, Darmstadt, Germany,), rabbit anti-Cre (1:1500, #257 003, Synaptic Systems, Goettingen, Germany) [18,54,55,56], Mouse anti-calbindin (1:1500, #300, Swant, Burgdorf, Switzerland ) [57,58,59,60,61,62,63,64,65,66], goat anti-Iba1 (1:300, #NB 100-1028, Novus Biologicals, Abingdon, UK) [67,68,69,70,71], rabbit anti-Iba1 (1:500, #ab178847, Abcam, Cambridge, UK) [67,68,69,70,71], mouse anti-NeuN (1:500, #MAB377, Millipore, Merck, Darmstadt, Germany) [72,73,74,75], rabbit anti-TH (1:1000, #P40101-150, Pelfreez, Rogers, AR, USA) [76,77,78], and chicken anti-TH (1:500, #ab76442, Abcam, Cambridge, UK) [76,77,78] (see Appendix A for additional information). The secondary antibodies used are donkey anti-goat-649 (Jackson Immunoresearch, Ely, UK, #705-496-147), donkey anti-chicken-488 (Jackson Immunoresearch, #703-545-155), donkey anti-chicken-Cy3 (Jackson Immunoresearch, Ely, UK, #703-166-155), donkey anti-mouse-Cy3 (Jackson Immunoresearch, Ely, UK, #715-166-150), donkey anti-mouse IgG1-647 (R&D Systems, Minneapolis, MN USA, #IC002R), donkey anti-rabbit-488 (Jackson Immunoresearch, Ely, UK, #711-546-152), donkey anti-rabbit-Cy5 (Jackson Immunoresearch, Ely, UK, #711-175-152), and donkey anti-rabbit-647 (Jackson Immunoresearch, Ely, UK, #711-606-152).

For Sema3A immunofluorescence, slices were first incubated with chondroitinase ABC (ChABC, 0.1 U/mL; Sigma-Aldrich, #C3667) for 2 h at 37 °C in ChABC buffer (0.1 M Tris–HCl, 0.03 M sodium acetate, pH 8.0), to partially digest the PNN as this step significantly improves the signal intensity of Sema3A in the PNN [4]. Then, slices were processed with a tyramide signal amplification biotin system (Akoya Biosciences, Marlborough, MA, USA, #NEL700A001KT) following the manufacturer’s protocol. Next, slices were incubated overnight at 4 °C with a goat anti-Sema3A antibody (C-17, Santa Cruz Biotechnology, Dallas, TX, USA, #sc-1146, 1:500) [4]. The following day, slices were incubated with a secondary biotinylated horse anti-goat IgG antibody (Vector Laboratories, Newark, CA USA, #BA-9500-1.5, 1:200) for 30 min RT, followed by an incubation with Streptavidin-HRP (1:150, TSA kit) for 30 min RT. Slices were then incubated with biotinylated-tyramide (1:60, TSA kit) for 3 min RT. Lastly, slices were incubated with streptavidin-Cy3 (Jackson Immunoresearch, #016-160-084) diluted in PBS for 30 min RT. Staining for other markers was performed on the same tissue following the classical staining protocol described above. After each step of Sema3A staining, slices were washed for 10 min in PBS three times. Finally, the sections were mounted in Mowiol (Calbiochem). For each immunohistochemical reaction, slices from all experimental conditions were processed together.

### 4.8. Imaging

Images were acquired using a confocal microscope (SP5 and SP8, Leica Microsystems) or a ZEISS Axioscan 7 and taken at a fixed laser intensity gain and offset for each analysis. Confocal images were taken using a 20× or 40× objective, depending on the analysis, at a resolution of 2048 × 2048 dpi and 100 Hz speed. Axioscan images were taken using a 10× objective at a resolution of 0.65 µm^2^ pixel size. Quantitative evaluations were made using Fiji (version 1.54f) or QuPath (version 0.4.3) software in a blind manner.

### 4.9. Quantification of Viral Transduction

To measure the transduction efficiency of AAV-PHP.eB-L7-GFP in PCs (Figure 1), the percentage of GFP^+^ PCs, stained by anti-calbindin antibodies, was evaluated in the vermis (3 slices/mouse) or the cerebellar hemisphere (3 slices/mouse) in 3 mice. Namely, the number of calbindin^+^ and GFP^+^/calbindin^+^ neurons in a region of interest (ROI) drawn around the PC layer were automatically counted using QuPath. To measure the transduction efficiency of AAV5-CAG-GFP (Figure 2) and AAV1-CAG-GFP (S.1) in the ChP, the percentage of Sema3A^+^ ChP cells expressing GFP and the percentage of GFP^+^ ChP cells expressing Sema3A in the lateral ventricle (4 slices/mouse) and in the 4th ventricle (4 slices/mouse) was evaluated for each virus (AAV5 mice: N = 3; AAV1 mice: N = 4). Namely, the number of GFP^+^, Sema3A^+^, and GFP^+^/Sema3A^+^ cells was manually counted using Fiji software (version 1.54f).

### 4.10. Quantification of Sema3A-Positive PNNs

The intensity of Sema3A staining around PNNs in the anterior and posterior parts of the interpositus nucleus and the barrel field of the somatosensory cortex was measured by drawing ROIs enwrapping each Sema3A^+^ PNN (PC KO group) or by using the method published by Slaker et al. (2016) [79] (ChP KO group and global KO group), in both cases by using Fiji software (version 1.54f) [(PC GFP mice: N = 5 (4 slices/mouse); PC KO mice: N = 3 (4 slices/mouse); ChP GFP mice: N = 3 (3 slices/mouse); ChP KO mice: N = 3 (3 slices/mouse); global GFP mice: N = 6 (4 slices/mouse); global KO mice: N = 4 (4 slices/mouse)]. For each slice, the intensity of the background, obtained by drawing a single ROI outside the Sema3A^+^ PNNs, was subtracted from the PNN intensities of the same slice.

### 4.11. Quantification of Iba1 Staining

The intensity of Iba1 staining was measured by drawing an ROI around the interpositus nucleus, the ventral horn of the cervical spinal cord, and the substantia nigra pars compacta, by using Qupath software (version 0.4.3) (3 slices/mouse; global GFP mice: N = 6; global KO mice: N = 6 for the brain analysis, N = 5 for the spinal cord analysis; CD-1 uninjected mice: N = 4; CD-1 Cre mice: N = 5 for the brain analysis, N = 3 for the spinal cord analysis). For each slice, the background intensity was obtained by drawing an ROI between microglial cells, and the average of the background values of all slices was calculated for each mouse. This average was subtracted from the Iba1 intensity of the same mouse. For clarity purposes, the data of Cre-mice were then normalized with their respective control.

### 4.12. Quantification of NeuN-Positive Neurons

The density of NeuN^+^ cells was measured by drawing an ROI around the interpositus nucleus and the ventral horn of the cervical spinal cord to obtain the surface size and count the number of NeuN^+^ cells in those areas (manually in the interpositus nucleus and automatically in the spinal cord), by using Qupath software (version 0.4.3) (3 slices/mouse; global GFP mice: N = 6; global KO mice: N = 6 for the brain analysis, N = 5 for the spinal cord analysis; CD-1 uninjected mice: N = 4; CD-1 Cre mice: N = 5). For each slice, the background intensity was obtained by drawing an ROI between NeuN^+^ cells, and the average of the background values of all slices was calculated for each mouse. The threshold for NeuN^+^ specific signal was set at 2 times the averaged background intensity for the global KO group and 1.5 times for the CD-1 group.

### 4.13. Quantification of TH-Positive Neurons

The density of TH^+^ cells was measured by drawing a ROI around the substantia nigra pars compacta to obtain the surface size and manually count the number of TH^+^ cells using Qupath software (version 0.4.3) (3 slices/mouse; global GFP mice: N = 6; global KO mice: N = 6). For each slice, the background intensity was obtained by drawing an ROI in an area without TH^+^ cells, and the average of the background values of all slices was calculated for each mouse. The threshold for TH^+^ specific signal was set at 2 times the average background intensity.

### 4.14. Statistics

Statistics were performed using GraphPad Prism 9 and 10 (GraphPad Software Inc., version 10.0.3, La Jolla, CA, USA). The normality of the data was tested using the Shapiro-Wilk test and if normality was violated, non-parametric tests were used. The rotarod results were analyzed by using a two-way ANOVA with repeated measures and a Šídák multiple comparisons test. Conditions and sessions/days were selected as fixed effects and the latency to fall as dependent variables. A Mann–Whitney U test was used to compare the frequency distribution of Sema3A^+^ PNNs. An unpaired *t*-test was used to compare the Iba1 intensity, NeuN density, and TH density. Data are reported as mean± standard deviation (SD) and considered significantly different when *p* < 0.05 (*), *p* < 0.01 (**), *p* < 0.001 (***), and *p* < 0.0001 (****). Statistical power was calculated using G-power software (version 3.1.9.7).

## Figures and Tables

**Figure 1 ijms-26-00819-f001:**
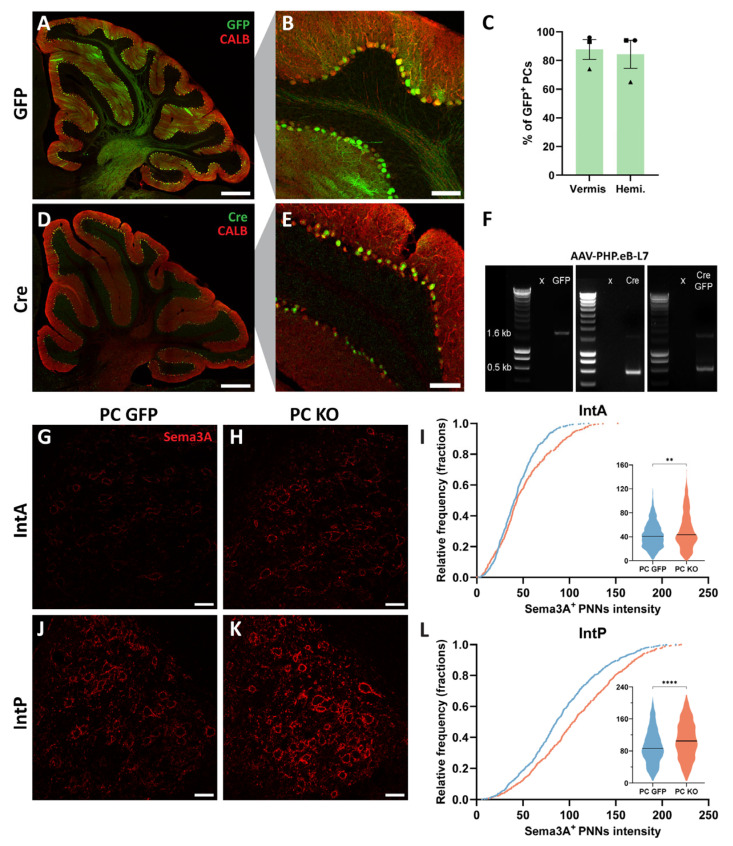
Knocking out *Sema3A* in PCs does not induce a decrease in Sema3A content in PNNs around cerebellar nuclei neurons. (**A**–**E**) Strong GFP (**A**,**B**) or Cre expression (**D**,**E**) is detected in the majority of PCs after intravenous injection in *Sema3A^fl/fl^* mice with AAV-PHP.eB-L7-GFP or Cre. (**C**) Quantification of the percentage of PCs expressing GFP in the vermis and in the cerebellar hemispheres (N = 3 mice, each mouse is represented by a shape). (**F**) PCR for *Sema3A* showing a band of 1.6 kb, which corresponds to the intact *Sema3A* gene, in PCs of *Sema3A^fl/fl^* mice injected with AAV-PHP.eB-L7-GFP, -Cre, or -CreGFP and a band of 0.5 kb, corresponding to the excised *Sema3A* gene, in the Cre conditions. (**G**–**L**) Sema3A intensity levels are increased in the IntA (**G**–**I**) and IntP (**J**–**L**) of PC KO mice when compared to PC GFP mice. (**I**,**L**) Cumulative frequency and violin plots of the intensity of individual Sema3A^+^ PNNs in the IntA and IntP of PC GFP mice (blue) and PC KO mice (red) (IntA, PC GFP: n = 651, PC KO: n = 457; IntP, PC GFP: n = 811, PC KO: n = 529). The median is represented by a black line. Scale bar: (**A**,**D**): 400 µm; (**B**,**E**): 100 µm; (**G**,**H**,**J**,**K**): 40 µm. ** *p* < 0.01, **** *p* < 0.0001.

**Figure 2 ijms-26-00819-f002:**
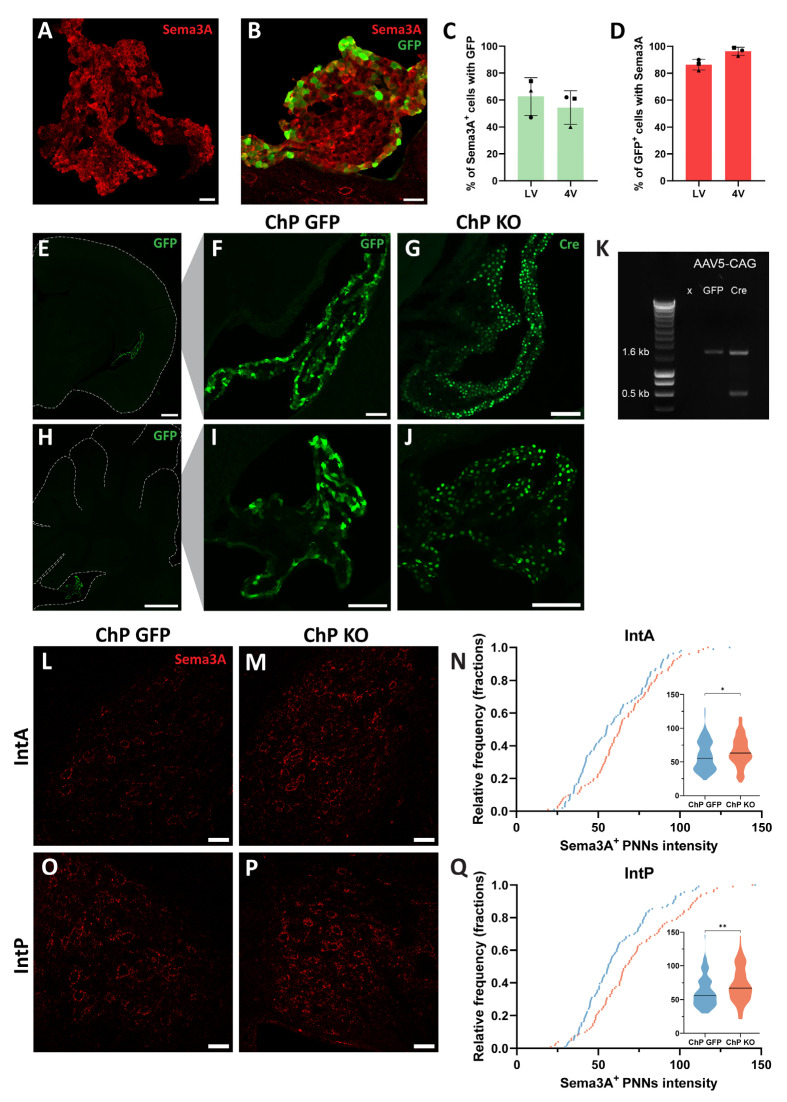
Knocking-out *Sema3A* in the ChP does not induce a decrease in Sema3A content in PNNs around cerebellar nuclei neurons. (**A**) Immunostaining reveals Sema3A expression in the ChP. (**B**) GFP^+^ cells in the ChP following injection of AAV5-CAG-GFP in the lateral ventricle. Several ChP cells express Sema3A. (**C**,**D**) Around 60% of Sema3A^+^ cells in the ChP express GFP (**C**) and around 90% of GFP^+^ cells are Sema3A^+^ (N = 3) (**D**) when mice are injected with AAV5-CAG-GFP. (**E**–**J**) Examples of the specific transduction of the ChP in the lateral ventricle (**E**) and the IV ventricle (**H**) following AAV5-CAG-GFP injection. The hemi-forebrain in coronal slice (**E**) and the cerebellum in sagittal slice (**H**) is outlined in dashed lines. (**F**,**G**,**I**,**J**) High magnification pictures showing GFP (**F**,**I**) or Cre expression (**G**,**J**) in the lateral ventricle (**F**,**G**) and the IV ventricle (**I**,**J**). (**K**) PCR for *Sema3A* on DNA extracted from the ChP of mice injected with AAV5-CAG-GFP or AAV5-CAG-Cre. A band of 0.5 kb is apparent in the Cre condition. (**L**–**Q**) Sema3A levels around neurons of the IntA (**L**–**N**) and IntP (**O**–**Q**) are increased in ChP KO mice. (**N**,**Q**) Cumulative frequency and violin plots of the intensity of individual Sema3A^+^ PNNs in the IntA and IntP of ChP GFP mice (blue) and ChP KO mice (red) (IntA, ChP GFP: n = 109, ChP KO: n = 102; IntP, ChP GFP: n = 116, ChP KO: n = 101). The median is represented by a black line. Scale bar: (**A**,**B**,**L**,**M**,**O**,**P**): 40 µm; (**E**,**H**): 600 µm; (**F**,**G**,**I**,**J**): 100 µm. * *p* < 0.05, ** *p* < 0.01.

**Figure 3 ijms-26-00819-f003:**
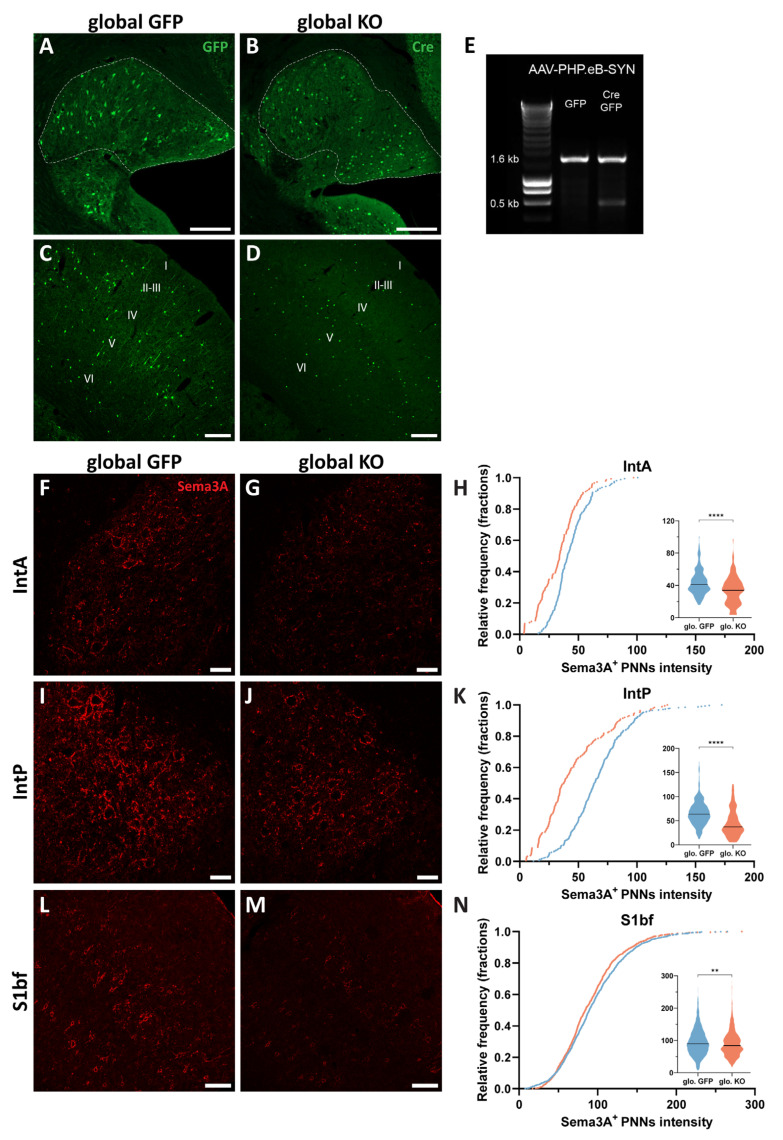
Globally knocking out *Sema3A* decreases Sema3A content in PNNs. (**A**–**D**) GFP (**A**,**C**) and Cre expressing cells (**B**,**D**) in the Int ((**A**,**B**); dashed line) and the S1bf ((**C**,**D**); cortical layers are indicated by roman numbers) after injection of AAV5-CAG-Cre, AAV-PHP.eB-L7-Cre, and AAV-PHP.eB-SYN-CreGFP in *Sema3A^fl/fl^* mice. (**E**) PCR for *Sema3A* on DNA extracted from the somatosensory cortex of global GFP and global KO mice. A correct excision in the *Sema3A* gene occurred in global KO mice. (**F**–**N**) Sema3A immunoreactivity is decreased in the IntA (**F**–**H**), the IntP (**I**–**K**), and the S1bf (**L**–**N**) of global KO mice. (**H**,**K**,**N**) Cumulative frequency and violin plots of the intensity of individual Sema3A^+^ PNNs in global (glo.) GFP mice (blue) and global KO mice (red) (IntA, global GFP: n = 162, global KO: n = 148; IntP, global GFP: n = 335, global KO: n = 187; S1bf, global GFP: n = 1660, global KO: n = 1074). The median is represented by a black line. Scale bar: (**A**–**D**): 200 µm; (**F**,**G**,**I**,**J**): 40 µm; (**L**,**M**): 100 µm. ** *p* < 0.01, **** *p* < 0.0001.

**Figure 4 ijms-26-00819-f004:**
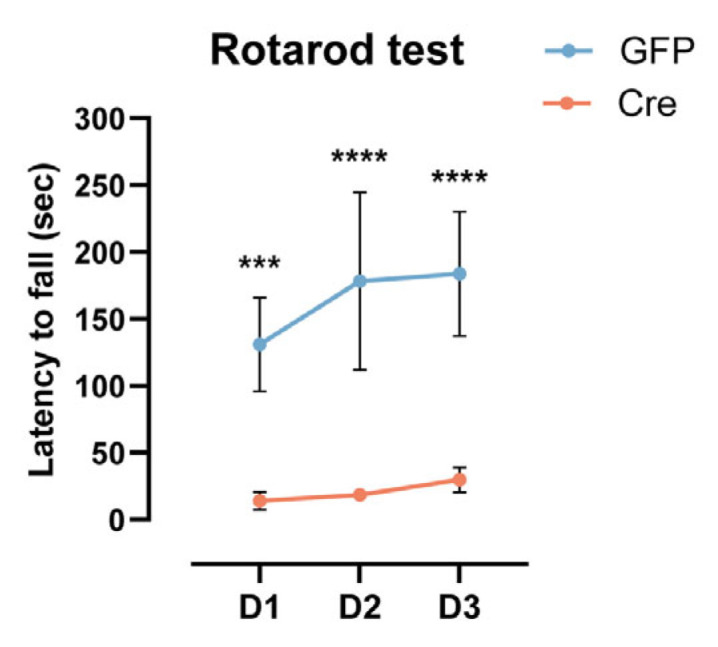
Cre expression throughout the CNS leads to motor deficits. Rotarod performance of global GFP mice (blue line) and global KO mice (red line). The latency to fall (seconds) is measured on three consecutive days (D) (global GFP: N = 6, global KO: N = 4). *** *p* < 0.001, **** *p* < 0.0001.

**Figure 5 ijms-26-00819-f005:**
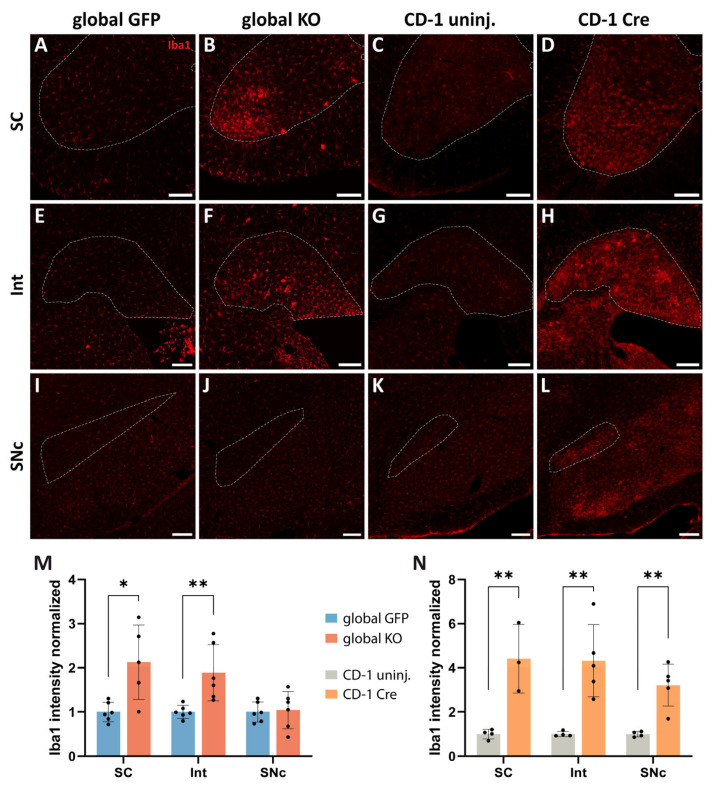
Cre expression throughout the CNS leads to neuroinflammation. (**A**–**D**,**M**,**N**) Increase in Iba1 immunoreactivity in the SC ventral horn of global KO mice and CD-1 Cre mice when compared to global GFP and CD-1 uninjected (uninj.) mice, respectively (global GFP: N = 6, global KO: N = 5; CD-1 uninjected N = 4, CD-1 Cre N = 3). (**E**–**H**,**M**,**N**) Increase in Iba1 immunoreactivity in the Int of global KO mice and CD-1 Cre mice when compared to global GFP and CD-1 uninjected mice, respectively (global GFP: N = 6, global KO: N = 6; CD-1 uninjected N = 4, CD-1 Cre N = 5). (**I**,**J**,**M**) No change in Iba1 intensity in the SNc of global KO mice when compared to global GFP mice (global GFP: n = 6, global KO: n = 6). (**K**,**L**,**N**) Increase in Iba1 levels in the SNc of CD1 Cre injected mice when compared to CD-1 uninjected mice (CD-1 uninjected N = 4, CD-1 Cre N = 5). Regions of interest are outlined by dashed lines. Scale bar: (**A**–**L**): 150 µm. * *p* < 0.05, ** *p* < 0.01.

**Figure 6 ijms-26-00819-f006:**
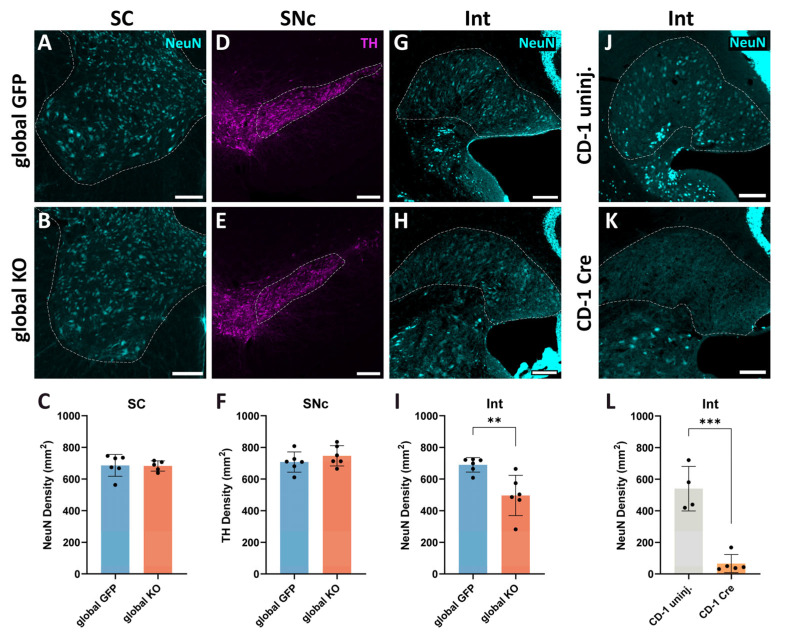
Cre expression throughout the CNS leads to neurodegeneration in the Int. (**A**–**C**) No change in NeuN^+^ neuronal density is apparent in the SC ventral horn of global KO mice when compared to global GFP mice (global GFP: N = 6, global KO: N = 5). (**D**–**F**) No change in TH^+^ neuronal density is apparent in the SNc of global KO mice when compared to global GFP mice (global GFP: N = 6, global KO: N = 6). (**G**–**L**) NeuN^+^ density is decreased in the Int of global KO mice when compared to global GFP mice (global GFP: N = 6, global KO: N = 6) (**G**–**I**) as well as in the Int of CD-1 Cre mice when compared to CD-1 uninjected (uninj.) mice (CD-1 uninjected: N = 4, CD-1 Cre: N = 5) (**J**–**L**). Regions of interest are outlined by dashed lines. Scale bar: (**A**,**B**,**G**,**H**,**J**,**K**): 150 µm; (**D**,**E**): 200 µm. ** *p* < 0.01, *** *p* < 0.001.

## Data Availability

Data not contained within this article or Appendix A are available upon request.

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
