# Peer review of "A Study on Potential Sources of Perineuronal Net-Associated Sema3A in Cerebellar Nuclei Reveals Toxicity of Non-Invasive AAV-Mediated Cre Expression in the Central Nervous System"

_ijms, 2025, doi:10.3390/ijms26020819_

Round 1
Reviewer 1 Report (Previous Reviewer 1)
Comments and Suggestions for Authors
The authors put an effort in revising their manuscript and addressing issues raised previously. The paper was improved.
Author Response
Please see the attachment.

Reviewer 2 Report (Previous Reviewer 2)
Comments and Suggestions for Authors
It is regrettable that the authors have not undertaken a comprehensive revision of the manuscript. The evaluation of antibody specificity through the comparison of reaction patterns with the data sheet provided by the manufacturer constitutes an inaccuracy. It is imperative to conduct independent positive, negative, or preadsorption control tests.
Round 2
Reviewer 2 Report (Previous Reviewer 2)
Comments and Suggestions for Authors
Now the authors have properly addressed my comments
This manuscript is a resubmission of an earlier submission. The following is a list of the peer review reports and author responses from that submission.
Round 1
Reviewer 1 Report
Comments and Suggestions for Authors
Gimenez et al. reported potential sources of perineuronal net (PNN)-associated Sema3A in cerebellar nuclei and demonstrated that non-invasive, viral-mediated Cre expression throughout the CNS can lead to toxicity. This paper is interesting, as it shows that knocking out Sema3A across the CNS reduces PNN-associated Sema3A. However, the study also revealed motor deficits, microgliosis, and neurodegeneration, which were attributed to Cre toxicity. Nonetheless, there are several issues that need to be addressed to further improve the manuscript.
1. The authors claimed that this study is the first attempt to unravel the cellular sources of PNN-associated Sema3A. They clearly demonstrated that knocking out Sema3A in PCs or the choroid plexus alone was insufficient to reduce the levels of PNN-associated Sema3A. In contrast, knocking out Sema3A across the entire CNS led to a reduction in PNN-associated Sema3A. However, the precise cellular source of PNN-associated Sema3A in cerebellar nuclei remains unclear. The authors should address this unresolved question.
2. What is the function of PNN-associated Sema3A? What is its neurobiological significance? Also, does knock out of Sema3A throughout the CNS alter the function and morphology of PNNs in the CNS?
3. An investigation into how selectively knocking out Sema3A in neurons, excluding PCs, affects the Sema3A levels in PNNs within the CNS should be conducted.
4. Are motor related area neurons more susceptible to Cre toxicity?
5. For intravenous AAV delivery, AAV-PHP.eB-SYN-Cre GFP (1.2 × 10¹² GC/mouse) was injected via the tail vein. Have you confirmed whether an AAV vector at this dosage can cross the BBB and achieve widespread distribution throughout the CNS?
Reviewer 2 Report
Comments and Suggestions for Authors
The objective of the present study is to determine the cellular localisation of one of the most important neuronal signalling molecules, Semaphorin 3A, in the perineuronal nets of the cell nuclei of the central nervous system (CNS). The research is novel and meticulously planned, employing a comprehensive range of contemporary molecular biology tools. The discussion provides a comprehensive contextualisation of the present study in relation to other discoveries in the field. Nevertheless, the work contains certain shortcomings that require clarification prior to publication.
Specific comments are provided below:
Please clarify the number of animals used in the study, as stated on line 402.
Line 416: The term "expression" should be used exclusively to refer to genes.
Line 436: Please state the manufacturers of the preparations (Rimadyl or Temgesic).
Line 453 and other chapters: Please provide justification for the low N number (sometimes 3 and 4).
Line 466: Please provide details on the design and validation of the primers.
Line 471: Specify the dose of pentobarbital and the manufacturer.
Line 514: Explain how the specificity of the primary antibodies was verified. The inclusion of positive and negative controls (preadsorption tests) is essential.